# EQuARX: Efficient Quantized AllReduce in XLA for Distributed Machine Learning Acceleration

Ibrahim Ahmed, Clemens Schaefer, Gil Tabak, Denis Vnukov, Zenong Zhang, Felix chern, Anatoliy Yevtushenko, Andy Davis
{ibahmed, cjsschaefer, tabakg, vnukov, zenong, fchern, anatoliyy, andydavis}@google.com
Google

*Abstract*—While Large Language Models (LLMs) have become highly influential, their enormous scale presents significant deployment challenges. Efficiently serving these models typically requires distributing them across numerous accelerator devices, which introduces substantial performance overhead from inter-device communication (collectives). While model quantization has been widely adopted to reduce the memory and compute requirements of LLM weights and activations with minimal quality impact, applying quantization directly to collectives like AllReduce is inherently difficult due to the inter-device summation involved, which can lead to numerical instability or significant error accumulation. In this work, we present a native dynamic block-wise efficient quantized AllReduce within the XLA compiler for TPUs (EQuARX). By using TPU-friendly quantization and deep pipelining of communication and compute, EQuARX with int8 precision achieves a $1.8\times$ speedup over baseline BF16 AllReduce across various network topologies. Furthermore, EQuARX accelerates the prefill stage of Gemma 3 27B by $1.25\times$ and Gemma 3 12B by $1.1\times$, respectively, with small to negligible impact on quality.

## I. INTRODUCTION

Large Language Models (LLMs) have demonstrated remarkable capabilities, driving significant advancements across diverse fields such as natural language understanding, generation, translation, and reasoning [2], [28]. Their success has fueled a trend towards models of ever-increasing scale, often following predictable scaling laws [19]. Training and serving these LLMs frequently exceeds the memory and compute capacity of a single accelerator device. Consequently, distributing the model (sharding) across clusters of many interconnected devices such as GPUs and TPUs has become standard practice for both training and inference [23], [28] to meet a target latency. While distribution allows the model to fit in memory and reduces latency, it introduces inter-device communication.

Different sharding strategies, such as model, pipeline, or data parallelism, introduce different kinds of collectives between the devices [10], [26], [27]. A ubiquitous and often performance-critical example is the AllReduce collective. AllReduce is fundamental for averaging calculated gradients across data-parallel workers during training [7], and it is also used to aggregate partial results when model parallelism sharding is used in both training and inference [26]. Although the communication overhead from AllReduce impacts performance in both scenarios, its effect is often more problematic during the serving/inference. Unlike in training, where the

AllReduce for gradient synchronization can sometimes be effectively overlapped with backward pass computation [22], AllReduce operations within model-parallel inference workloads frequently reside directly on the execution critical path [23]. Consequently, devices are forced to stall while waiting for these AllReduce operations to complete, which directly reduces per-device compute efficiency.

The importance of collective performance has driven hardware vendors to build high-bandwidth chip-to-chip network fabrics, such as Google's Inter-Chip Interconnect (ICI) for TPUs [17] and NVIDIA's NVLink/NVSwitch for GPUs [20]. Maximizing the benefit of these fast networks also requires intelligent software strategies that utilize the available bandwidth efficiently. One software-based optimization that has shown tremendous promise across machine learning (ML) workloads is quantization. Numerous studies have successfully applied quantization to model weights and activation. These efforts resulted in significant reductions in memory footprint and computational demands, often while maintaining high model quality and task performance across various domains [6], [11]. While naively quantizing tensors before performing an AllReduce would reduce the amount of data sent through the network, it can easily lead to numerical overflow if intermediate sums exceed the range of the low-precision format, or cause the accumulation of unacceptable error across the many participants, thereby degrading the model quality.

To reduce the overhead of AllReduce while maintaining model accuracy, we designed a highly efficient quantized AllReduce optimized for TPUs. Our quantized AllReduce includes a dynamic block-wise quantization/dequantization scheme performed concurrently with the reduction operations within the AllReduce collective. This approach significantly mitigates the quantization-induced errors that impact naive methods. Both quantization/dequantization and the collective algorithms were co-designed to extract maximum performance from the TPU compute units and the ICI network. We have implemented our efficient quantized AllReduce (EQuARX) as a native operation within the XLA compiler framework, making it seamlessly accessible to users through high-level interfaces like JAX. Our evaluations demonstrate the effectiveness of this design: EQuARX successfully hides most of the computational overhead associated with quantization and dequantization, achieving approximately 90% of the theoretical speedup expected from halving the volume of transmitted

data. We show that integrating EQuARX in Gemma 3 results in speeding up the prefill stage of the 27B model by 1.25X and the 12B model by 1.1X, with small to negligible accuracy drops.

## II. RELATED WORK AND BACKGROUND

The need for software-level solutions to accelerate communication in distributed machine learning workloads has been recognized over the last few years. For example, ZeRO++ [29] introduced several communication reduction techniques using quantization, including block-wise int8 quantization for all-gathers. For gradient averaging, they only need to perform a reduce-scatter (due to their chosen sharding strategy). They proposed decomposing it into an all-to-all collective followed by a local reduction. This decomposition allowed them to quantize gradients before the all-to-all and dequantize the output of the all-to-all before performing the local reduction. Unfortunately, implementing a reduce-scatter as an all-to-all followed by a local reduction results in underutilizing the network bandwidth significantly as opposed to the ring/bucket-based [12] algorithms which are proven to be bandwidth optimal [3] on torus topologies. Building on ZeRO++, QSDP [21] supports weight and gradient quantization; it uses randomized rounding for gradients and 'random shifts' so that the resulting weight estimate is unbiased. More recently, the work in SDP4Bit [13] compresses weight differences instead of weights directly, and uses a 2-level hierarchical scheme with different precision (and a Hadamard transform) for gradient compression.

gZCCL [9] developed a framework that supports compression-aware collectives. Their proposed algorithm uses recursive doubling with many optimizations to improve GPU utilization during compression. Most of the optimizations in gZZCL are specific to GPU hardware, GPU topology and programming model. In this work, we focus on TPUs, which have a different architecture and network topology.

TPUs [15] are custom machine learning (ML) acceleration chips built by Google that are widely adopted for training and inference of ML workloads. At its heart, the TPU has a powerful systolic array matrix multiplication unit [18]. TPUs use 2D vector registers and perform general computations on a vector processing unit (VPU) that operates on 2D operands of shape 8x128. TPUs are connected together through a high-speed inter-chip interconnect to form a 2-D or 3-D mesh or torus topologies (e.g. 2x2 or 4x4x4) [17].

## III. EFFICIENT QUANTIZED ALLREDUCE IN XLA

AllReduce can be executed as a reduce-scatter followed by an all-gather while still ensuring optimal network bandwidth utilization [3]. Since all-gather is a communication-only collective it is simpler to quantize. In this Section, we will focus on the different optimizations we designed to ensure that we can efficiently quantize the reduce-scatter stage.

Fig. 1 shows the three steps of a ring-based reduce-scatter algorithm on 4 TPUs forming a 1X4 torus. Initially, each TPU divides its input tensor (e.g. A) into 4 shards ($A_*$). At the first

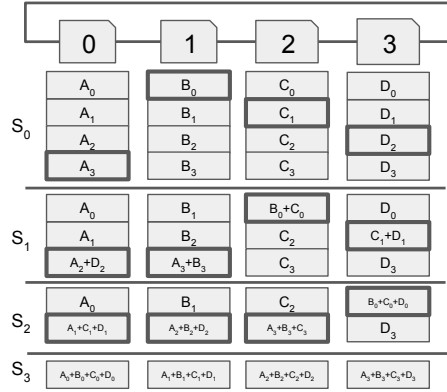

Fig. 1: Three iterations of a ring-based reduce-scatter algorithm on 4 TPUs.

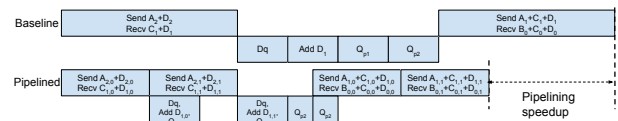

Fig. 2: Baseline and pipelined execution timeline of a single iteration of quantized reduce-scatter.

step, each TPU sends a shard to its neighbor. In the remaining steps, each TPU sums the received shard with its local counter part and then sends the partial result to its neighbor. After N-1 steps (where N is the number of TPUs in the ring), each TPU would have a shard of the final output as shown in Fig. 1.

### A. Deep Pipelining

To accelerate the all-reduce collective, we want to reduce the amount of data sent between devices by quantizing the data to a lower precision (e.g. int8 while the original tensor uses BF16). Since each step of the reduce-scatter involves an addition, we cannot simply quantize the input tensor and perform all additions in the lower precision without accumulating large errors and risking overflows. Instead, we quantize the shard, send the quantized shard and the corresponding metadata (i.e. scale factors), dequantize the received shard (Dq) to higher precision (e.g. FP32), and perform the addition. These steps are repeated for the N-1 iterations of the quantized reduce-scatter. While these steps allow us to safely reduce the data sent through the network, it introduces more compute that could result in under utilizing the network bandwidth.

Fig. 2 shows the baseline and our deeply pipelined execution time of a single iteration of the quantized reduce-scatter algorithm. We distinctively divide the quantization step into two phases as we need to iterate through the shard twice to quantize it. In the first iteration ($Q_{p1}$), we calculate metadata, and in the second iteration ($Q_{p2}$) we use this metadata to scale the data. As can be seen from Fig. 2, the overhead introduced by quantization and dequantization results in large bubbles during which the network is idle. To reduce the overhead of these bubbles, we divide each shard into *u*

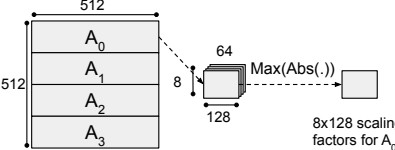

Fig. 3: 1024 scale factors for each shard of tensor A (i.e. one scale per 64 entries).

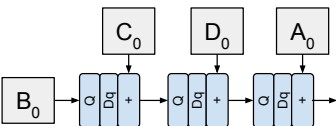

Fig. 4: Full-loop adder chain to calculate shard 0 result.

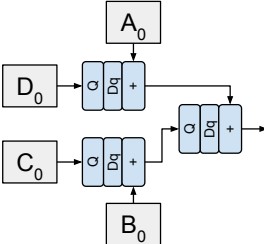

Fig. 5: Semi-loop adder chain to calculate shard 0 result.

microshards and pipeline the communication and compute of the different microshards when possible. In this example, $u$ is two (where $A_{i,j}$ is the $j^{th}$ microshard of the $i^{th}$ shard of A) and we are able to start dequantizing the first microshard while receiving the second microshard. We ensure that at each reduce-scatter iteration we first send the metadata before sending any microshards, which allows us to immediately start the dequantization process. Moreover, since there is no data dependency between the dequantization, addition and the first quantization phase ($Q_{p1}$) of different microshards, we pipeline all these operations together. However, to get the shard-wide metadata (e.g. maximum absolute value in a shard) we cannot start the second quantization phase ($Q_{p2}$) without completing the first phase of all microshards. As depicted in the figure, this deep pipelining allows us to significantly reduce the quantization overhead significantly and increase the network bandwidth utilization.

### B. Block-wise VPU-friendly Quantization

To quantize data in a tensor from a higher precision representation (e.g. BF16) to a lower precision (e.g. int8), we need to calculate some metadata to capture the range of the original data. For symmetric/scale quantization [30], we perform a maximum reduction to to determine the appropriate scale factor. Instead of calculating a single scale factor for each shard, we calculate 1024 (8x128) scale factors for each shard. We picked 8x128 to match the native 2D register vectors used by the TPU. This change enables us to only use the TPU's VPU to perform the absolute and max operations across different 8x128 chunks of the input shard to identify the scale factors without needing any expensive reshapes or data transformation (to perform the reduction within a 8x128 chunk). Fig. 3 shows how we calculate the scale factors of a shard ($A_0$) from a tensor with shape (512, 512). We divide the shard into 64 8x128 chunks and we reduce the 64 chunks into a single chunk through an absolute and maximum operations. This VPU-friendly quantization not only accelerates the calculation of the scale factors, but it also results in a higher-quality quantization as we are now performing a block-wise quantization with a block size of 64 in this example.

To better control the block size (independently from the shard size), we introduce a second layer of data division. Each shard is first divided into $m$ minishards, and then each minishard is divided into the $u$ microshards. We calculate the 8x128 scale factors across a single minishard. Thus, the number of entries reduced over for a single scaling factor is entirely determined by $m$. In the example shown in Fig. 3 the

block-size would be $\frac{64}{m}$. This allows us to tune $m$ to optimize for the quality of the quantization, and tune both $m$ and $u$ to optimize for performance. Increasing $m$ and $u$ reduce the bubble time shown in Fig 2. However, increasing $m$ results in more metadata going through the network.

### C. Full-loop vs Semi-loop Ring

We implemented two variants of the ring-based reduce-scatter algorithm: full-loop and semi-loop. Fig. 1 describes the full-loop variant. As shown in Fig. 4, one potential problem with this variant is that it results in many (N-1 for a ring with N devices) steps of quantization and dequantization, which could increase the accumulated error.

The semi-loop variant utilizes both directions of the ring and it only performs N/2 iterations. Fig. 5 shows the more balanced adder tree generated by the semi-loop variant to calculate shard 0 result. In this example, in the first iteration device 3 sends its $0^{th}$ shard to device 0 and at the same time device 2 sends its $0^{th}$ shard to device 1. Both recipient devices perform a dequantization and addition. In the second iteration, device 1 sends the partial result to device 0 that performs the final dequantization and addition to get the final result. While the semi-loop variant does result in a more balanced adder tree and has fewer hops, it is not as bandwidth efficient as running two full-loop variants in opposite direction. The lower bound of the bandwidth component of the reduce-scatter time of the full-loop variant is $\frac{(N-1)D}{2NB}$, while it is $\frac{D}{2B}$ for the semi-loop variant. Where $N$ is the number of devices forming the ring, $D$ is the number of bytes of the input, $B$ is the per-direction bandwidth in bytes/sec. We added support for both to allow users to pick the variant that better suits their use case.

### D. All-gather

To fully quantize the all-reduce, EQuARX also supports quantizing the all-gather that comes after the reduce-scatter stage. When quantizing the all-gather stage, we add an extra quantize and dequantize pair to the adder chains shown in Fig. 4 and Fig. 5. We implemented EQuARX such that users

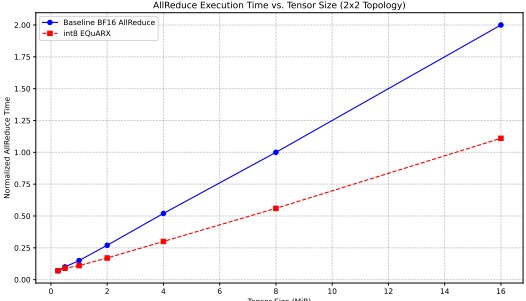

Fig. 6: Speedup of Full-loop EQuARX across different tensor sizes.

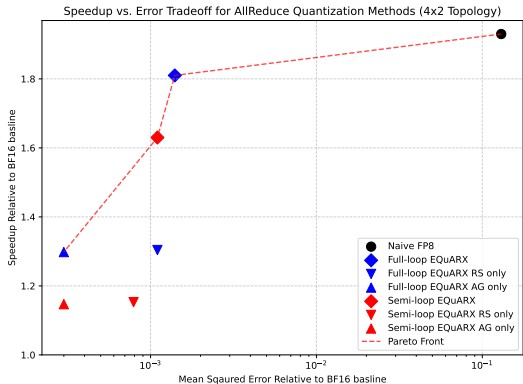

Fig. 7: Speedup vs error of EQuARX (with a block size of 64) relative to a BF16 baseline on a 4x2 topology on a tensor of shape (4096, 4096) with values sampled from $\mathcal{N}(0, 1)$.

can select whether to quantize the reduce-scatter (RS) stage, the all-gather (AG) stage or both. Quantizing both gives the best performance but potentially introduces more error than quantizing just a single stage.

## IV. MICROBENCHMARKS

EQuARX supports three data types for quantization: int8, FP8 (E4M3) and FP8 (E5M2), and it uses symmetric quantization. In this section, we will show results for int8 only. All results presented here are running on TPU v5e. Fig. 6 shows the normalized execution time of a BF16 baseline AllReduce and our full-loop int8 EQuARX with RS and AG stages quantized across different tensor sizes. For tensor sizes less than 2 MiB, both EQuARX and baseline AllReduce have similar performance as the execution time is dominated by the hop latency between TPUs and not the bandwidth of the links. For larger tensors, our int8 EQuARX execution time is ∼55% of the baseline BF16 AllReduce. This execution time is only 10% longer than the ideal execution time that would have been achieved by a 2X compression of the transferred data (no quantization and no metadata overhead). This result highlights that the optimizations implemented in EQuARX significantly reduce the overhead associated with quantization and dequantization of the partial results.

### A. EQuARX Speedup vs Error Trade-off

As explained earlier EQuARX supports different knobs that provide different performance vs error trade-offs. The first knob is whether to use full-loop or semi-loop EQuARX, and the second knob selects which stages of the AllReduce to quantize: reduce-scatter (RS) only, all-gather (AG) only, or both stages. To show an example of this trade-off we executed a baseline BF16 AllReduce on a tensor of shape (4096, 4096) running on a 4x2 TPU topology. For the sake of this test, we populated the tensor with a standard normal distribution (mean of 0 and a variance of 1). We also execute all the 6 flavors of our int8 EQuARX on the same input tensor. In addition to EQuARX, we also execute a naive FP8 (E5M2) AllReduce where we convert the BF16 input tensor to FP8 and then perform an AllReduce on the converted tensor (no quantization or scaling involved). We use this naive FP8 AllReduce as an estimate to the roofline performance improvement achieved by reducing the data traversing the network. The naive FP8 AllReduce is a roofline as it halves the data sent through the networks without adding any quantization/dequantization compute and it does not include the overhead of transferring the metadata associated with dynamic quantization.

Fig. 7 shows the speedup of the our different EQuARX flavors and the mean squared error (MSE) relative to the BF16 baseline. First observation is that while the naive FP8 AllReduce achieves 1.9X speedup, it results in a significant MSE of 0.13. While not shown here a naive FP8 could easily overflow resulting in corrupting the AllReduce completely. Our fastest EQuARX flavor results in a 1.8X speedup with only 0.0014 MSE (two orders of magnitude lower than the naive FP8 version while achieving 95% of the speedup).

As expected, the semi-loop EQuARX results in lower error compared to the full-loop EQuARX but at the expense of a reduced speedup. Semi-loop EQuARX (with both stages quantized) results in a 1.6X speedup with a lower MSE of 0.001. The EQuARX flavors with the lowest MSE are the ones that only quantize the AG stage; the full-loop EQuARX with only quantizing the AG stage results in a 1.3X speedup with the smallest MSE of 0.0003. These different knobs are exposed to users or automatic quantization tools to be able to squeeze the most performance while maintaining the end-to-end accuracy of the model within the acceptable range.

## V. EXPERIMENTS WITH GEMMA 3

To demonstrate EQuARX in a representative workload, we measure the performance and quality impact of EQuARX on serving Gemma 3 12B and 27B models running on TPU v5e, and using BF16 weights. The 27B model uses 16-way model parallelism on a 4x4 topology, while the 12B model uses 4-way model parallelism on a 2x2 topology. For both models, there are two AllReduce operations per layer due to the chosen sharding strategy. For LLM serving, there are two distinct stages: prefill and decode. Prefill processes the entire input context in a single iteration to primarily initialize KV caches and then the decode stage autoregressively generates one token per iteration (using the KV caches generated by the

TABLE I: Quality and prefill latency of Gemma 3 12B and 27B model. Performance benefits are most pronounced on the larger topology, where EQuARX offers up to $1.28\times$ prefill latency speedup over baseline AllReduce with a prefill sequence length of 2048.

| Benchmark | Examples | Metric | 12B Base | 12B Full-loop | | 12B Semi-loop | | 27B Base | 27B Full-loop | | 27B Semi-loop | |
|---|---|---|---|---|---|---|---|---|---|---|---|---|
| | | | Acc. % | Acc. % | Diff. | Acc. % | Diff. | Acc. % | Acc. % | Diff. | Acc. % | Diff. |
| MBPP [1] | 500 | 3-shot | 70.60 | 67.80 | $-2.80$ | 69.20 | $-1.40$ | 72.80 | 71.40 | $-1.40$ | 71.40 | $-1.40$ |
| HumanEval [4] | 164 | pass@1 | 78.66 | 78.66 | 0.00 | 79.27 | $+0.61$ | 83.54 | 82.93 | $-0.61$ | 82.93 | $-0.61$ |
| Boolq [5] | 3270 | 0-shot | 87.40 | 87.49 | $+0.09$ | 87.40 | 0.00 | 87.89 | 87.89 | 0.00 | 87.89 | 0.00 |
| HellaSwag [31] | 10042 | 10-shot | 81.41 | 80.93 | $-0.48$ | 81.36 | $-0.05$ | 83.26 | 81.46 | $-1.80$ | 81.46 | $-1.80$ |
| TriviaQA [16] | 7993 | 5-shot | 70.35 | 70.42 | $+0.07$ | 70.40 | $+0.05$ | 79.46 | 79.44 | $-0.02$ | 79.46 | 0.00 |
| Winogrande [25] | 1267 | 5-shot | 74.19 | 74.43 | $+0.24$ | 74.27 | $+0.08$ | 75.77 | 75.77 | 0.00 | 75.77 | 0.00 |
| AGIEval [32] | 2340 | 3-5-shot | 57.64 | 58.80 | $+1.16$ | 58.07 | $+0.43$ | 67.44 | 64.86 | $-2.58$ | 65.08 | $-2.36$ |
| MMLU [8] | 14042 | 5-shot | 72.00 | 71.96 | $-0.04$ | 71.92 | $-0.08$ | 77.47 | 77.12 | $-0.35$ | 77.14 | $-0.33$ |
| GPQA [24] | 448 | 5-shot | 30.36 | 29.02 | $-1.34$ | 31.03 | $+0.67$ | 34.60 | 37.95 | $+3.35$ | 34.15 | $-0.45$ |
| MedQA [14] | 500 | 5-shot | 68.00 | 65.40 | $-2.60$ | 65.60 | $-2.40$ | 75.20 | 73.40 | $-1.80$ | 75.40 | $+0.20$ |
| Prefill Length | | | Lat. (ms) | Lat. (ms) | Speedup | Lat. (ms) | Speedup | Lat. (ms) | Lat. (ms) | Speedup | Lat. (ms) | Speedup |
| 2048 | | | 116.17 | 105.67 | $1.10\times$ | 109.87 | $1.06\times$ | 111.44 | 88.06 | $1.27\times$ | 87.24 | $1.28\times$ |
| 4096 | | | 243.94 | 221.93 | $1.10\times$ | 230.37 | $1.06\times$ | 242.57 | 197.3 | $1.23\times$ | 194.87 | $1.25\times$ |
| 8192 | | | 604.12 | 555.23 | $1.09\times$ | 572.5 | $1.06\times$ | 550.36 | 456.68 | $1.21\times$ | 452.97 | $1.22\times$ |

prefill stage). As a result, the input to the AllReduce in the decode stage is much smaller than that of the prefill and so the AllReduce of the decode stage is mostly latency bound. Since EQuARX is beneficial for AllReduce with large inputs, we apply it only to the AllReduce of the prefill stage.

Table I shows the accuracy achieved across 10 different tasks for both models when using a baseline BF16 Allreduce, a full-loop EQuARX and a semi-loop EQuARX. The table also shows the prefill latency. EQuARX results in small to negligible accuracy drop across all tasks. For the 12B model, full-loop EQuARX results in a 1.1X speedup, while the semi-loop version results in a 1.06X speedup across a range of prefill lengths. For the 27B model, the semi-loop variant results in a 1.28X speedup for the 2048 prefill length. Since the AllReduce executed in the 27B runs on 16 devices the semi-loop and full-loop EQuARX perform similarly (unlike the 12B case).

While the largest percentage drops were found in MBPP and MedQA, these results were not statistically significant (using p=0.05) due to the relatively small number of examples in these dataset (both 500). While none of the 12B deviations were found to be statistically significant, for 27B the HellaSwag and AGIEval were (N=10042 and 2340, respectively). We also generally saw improvements using the semi-loop instead of the full-loop algorithm.

## VI. CONCLUSION

In this paper we introduced EQuARX to accelerate distributed machine learning workloads by speeding up AllReduce operations. EQuARX is native to XLA and supports a wide range of network topologies; it uses a dynamic block-wise quantization scheme to reduce the accumulated error throughout the collective and avoid risks of overflow. By co-designing EQuARX with TPU architecture in mind, we were able to minimize the overhead associated with quantization/dequantization. We applied EQuARX to Gemma 3 models where it yielded $1.27\times$ performance improvement to the prefill stage with small to negligible quality impact.

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
