# OpenReview forum: "EQuARX: Efficient Quantized AllReduce in XLA for Distributed Machine Learning Acceleration"
_iscaconf.org/ISCA/2025/Workshop/MLArchSys — MLArchSys 2025 Oral_

### Official Review · Reviewer_PwBM · 2025-05-17
**Effective technique for an important problem in optimizing collective communications**

**Confidence:** 3
**Rating:** 6

**Detailed Feedback And Questions For Authors:**

The paper focuses on a relevant and important problem in distributed training (and inference): the communication overhead of collective communications such as AllReduce. The proposed EQUARX technique introduces a novel dynamic block-wise quantization approach that addresses challenges in error accumulation when applying quantization for collective communication. The evaluations are thorough, and the paper is well-written and easy to follow.

A few areas where the paper can be strengthened:

*The evaluation and design are specifically focused on TPUs. While the paper claims that EQUARX supports a wide range of network topologies, the evaluation predominantly focuses on TPU torus configurations. How can the approach be generalized to other hardware architectures, and how can network configuration impact the performance of this technique?

*Elaborating on how the method scales with an increasing number of nodes in the collective communication would further strengthen the approach's applicability.

*Additional evaluations on the impact of different quantization precisions (e.g., int8, FP8) on performance and accuracy would strengthen the paper and enhance the insights.

**Top Reasons To Accept The Paper:**

* The paper focuses on a relevant and important problem on distributed training (and inference). Which is the communication overhead of collective communications such as AllReduce.

* The paper proposes a novel dynamic block-wise quantization approach for all reduce. which effectively addresses challenges in error accumulation when trying to apply quantization for collective communications

* The evaluation results are thorough through microbenchmarks and demonstrate the performance improvements (and minimal accuracy degradation) of the approach.

* The paper is well-written and easy to follow.

**Top Reasons To Reject The Paper:**

* The approach is specifically focused on TPU architectures and also only evaluated on TPUs

- While the paper claims that EQUARX 'supports a wide range of network topologies', the evaluation is predominantly focused on TPU torus configurations (e.g., 2x2, 4x2, and 4x4).  How can network impact the performance of this technique?

* There is minimal discussion on the scalability of the approach. How do the performance improvements shown in the evaluation section scale as the number of nodes involved in the collective communication increases?

---

### Official Review · Reviewer_4mZe · 2025-05-18

**Confidence:** 3
**Rating:** 6

**Detailed Feedback And Questions For Authors:**

-

**Top Reasons To Accept The Paper:**

It combines block-wise quantization, deep pipelining, and architecture-aware quantization (e.g., 8×128 VPU-friendly blocks) for efficient execution, showcasing strong co-design.

**Top Reasons To Reject The Paper:**

Will be nice to have more comprehensive evaluations across diverse models and sizes.

---

### Official Review · Reviewer_ZJ1v · 2025-05-19
**The paper proposes EQuARX, a block-wise quantized AllReduce implementation which pipelines and hierarchically shards data to reduce latency and with minimal loss of accuracy.**

**Confidence:** 4
**Rating:** 6

**Detailed Feedback And Questions For Authors:**

Thanks for submitting the paper.
Some minor suggestions below:

1. It would have been beneficial to show the sending of metadata in Fig 2 as well.

2. While the less number of shards shown in Fig2 showcases speedup, the benefit at higher number of shards might be negligible as one would have to wait for the Qp1 stage of all shards? That was unclear.

3. Some discussion on why we can't propose a similar architecture for GPUs or alternatively, why such a TPU-specific solution is needed would have been good to have.

**Top Reasons To Accept The Paper:**

1. The paper is well written and proposes quantized communication collectives with pipelining and blockwise sharding techniques which match the state-of-the-art quantization techniques today - thus maintaining parity.
2. The work enables large gains in the form of speedup with negligible accuracy loss.
3. The sharding and deep pipelining achieves close to theoretical speedup (FP8) thus mitigating any loss coming from quantization and dequantization (Fig7)

**Top Reasons To Reject The Paper:**

1. My main concern is that it is unclear why techniques which were proposed for GPUs cannot be ported to TPUs. The authors mention different architecture and network topology of TPUs - however it is not clear how it affects solutions such as gZCCL. gZCCL also deals with ring topologies and proposes similar solutions of pipelining for GPUs. The compute architecture difference seems to be unrelated to the problem at hand. Some more discussion about it would have been beneficial.
2. It would have been interesting to see how EQUARX would fare with quantized models which already have blockwise metadata for MLP and attention GEMMs. Some discussion about integrating the allgather/reduce with those techniques would have been interesting to understand. Basically, what would happen if your input was INT8 already with blockwise metadata to begin with - instead of BF16.

---

### Official Review · Reviewer_d75m · 2025-05-21
**Dynamic block-wise quantized AllReduce support in XLA**

**Confidence:** 2
**Rating:** 5

**Detailed Feedback And Questions For Authors:**

This paper presents an efficient implementation AllReduce in XLA for TPU. The operation supports flexible ring-based reducescatter algorithms (full-loop vs semi-loop) and quantization targets (both, AG,RS).
It is a good feature to add to XLA and is shown to accelerate LLM serving on TPU. It would be great to extend this work to support lower precision and more flexible mixed precision block-wise quantization schemes.

**Top Reasons To Accept The Paper:**

+ applies effective techniques to speed up AllReduce, such as blockwise int8 quantization and deep pipelining
+ achieves 1.8x speedup over BF16 in AllReduce with little impact on accuracy
+ easily accessible within in the XLA compiler framework

**Top Reasons To Reject The Paper:**

- the optimizations are very TPU specific
- limited novelty